# High-Temperature Magnesiothermic Reduction Enables HF-Free Synthesis of Porous Silicon with Enhanced Performance as Lithium-Ion Battery Anode

**DOI:** 10.3390/molecules27217486

**Published:** 2022-11-02

**Authors:** Xiuxia Zuo, Qinghua Yang, Yaolong He, Ya-Jun Cheng, Shanshan Yin, Jin Zhu, Peter Müller-Buschbaum, Yonggao Xia

**Affiliations:** 1Ningbo Institute of Materials Technology & Engineering, Chinese Academy of Sciences, 1219 Zhongguan West Rd., Ningbo 315201, China; 2Shanghai Institute of Applied Mathematics and Mechanics, School of Mechanics and Engineering Science, Shanghai University, Shanghai 200444, China; 3Department of Materials, University of Oxford, Parks Rd., Oxford OX1 3PH, UK; 4Physik-Department, Lehrstuhlfür Funtionelle Materielien, Technische Universität München, James-Franck-Strasse 1, 85748 Garching, Germany; 5Heinz Maier-Leibnitz Zentrum (MLZ), Technische Universität München, Lichtenbergstr. 1, 85748 Garching, Germany; 6Center of Materials Science and Optoelectronics Engineering, University of Chinese Academy of Sciences, 19A Yuquan Rd., Shijingshan District, Beijing 100049, China

**Keywords:** lithium-ion battery anode, porous silicon, HF-free, magnesiothermic reduction

## Abstract

Porous silicon-based anode materials have gained much interest because the porous structure can effectively accommodate volume changes and release mechanical stress, leading to improved cycling performance. Magnesiothermic reduction has emerged as an effective way to convert silica into porous silicon with a good electrochemical performance. However, corrosive HF etching is normally a mandatory step to improve the electrochemical performance of the as-synthesized silicon, which significantly increases the safety risk. This has become one of the major issues that impedes practical application of the magnesiothermic reduction synthesis of the porous silicon anode. Here, a facile HF-free method is reported to synthesize macro-/mesoporous silicon with good cyclic and rate performance by simply increasing the reduction temperature from 700 °C to 800 °C and 900 °C. The mechanism for the structure change resulting from the increased temperature is elaborated. A finite element simulation indicated that the 3D continuous structure formed by the magnesiothermic reduction at 800 °C and 900 °C could undertake the mechanical stress effectively and was responsible for an improved cyclic stability compared to the silicon synthesized at 700 °C.

## 1. Introduction

Lithium-ion batteries (LIBs) have been widely applied as dominant energy storage devices due to their high energy and power densities, long cycling life, and environmental benignity [1,2,3]. The performance of LIBs greatly depends on their electrodes. However, commercial graphite-based anodes have difficulty satisfying the high performance demanded due to their low theoretical capacity (372 mAh g^−1^) and serious safety concerns [4]. Silicon, which features a high theoretical specific capacity of about 4200 mAh g^−1^ and an appropriate working voltage of around 0.5 V (vs. Li/Li^+^), is viewed as one of the most promising candidates to replace graphite [5,6,7]. However, it suffers from poor cyclic stability due to the severe volume expansion (>300%) during the Li insertion/extraction process. Porous silicon has gained much interest because its pores can effectively accommodate the volume changes and release the mechanical stress, leading to improved cycling performance and rate performance [8,9,10].

Magnesiothermic reduction has emerged as an effective way to convert silica into porous silicon with a good electrochemical performance [11,12,13]. HF etching is normally a mandatory step to treat the as-synthesized silicon to remove residual silica, which is crucial to improve the electrochemical performance. However, the corrosive HF etching process significantly increases the safety risk and makes it environmentally unfriendly [14,15]. This has become one of the major issues that impedes the practical application of the magnesiothermic reduction synthesis of porous silicon anodes. Although few works have mentioned the use of HCl instead of an aggressive HF etchant, the potential mechanism for the structure change and the electrochemical responses using an increased temperature has not been elaborated.

Our recent work developed a new method to synthesize three-dimensional hierarchical macro-/mesoporous silicon from zero-dimensional Stöber silica particles using a magnesiothermic reduction process [16,17]. Significantly improved cyclic and rate performances were demonstrated by the porous silicon compared to the commercial nano-sized and micrometer-sized silicon particles. The macrosized pores effectively accommodated the volume expansion to alleviate the mechanical stress and the mesosized pores enhanced the electrochemical kinetics. With the coexistence of both macro- and mesosized pores, the reversible specific capacity was still retained at as much as 959 mAh g^−1^ after 300 cycles at 0.2 A g^−1^ with a mass loading density of 1.4 mg cm^−2^.

Despite the success of our previous work, HF treatment was still needed to remove residual silica and improve the electrochemical performance. In this work, a facile HF-free method to synthesize macro-/mesoporous silicon with good cyclic and rate performance was established by simply increasing the reduction temperature, resulting in a good electrochemical performance with reasonable mass loading densities. The potential mechanism for the structure change caused by the increased temperature was elaborated. A finite element simulation applied to gain insights into the mechanical response to lithiation of the silicon synthesized at different temperatures; the simulation revealed that the 3D continuous structure formed at 800 °C and 900 °C was beneficial to undertaking mechanical stress while the structure formed at 700 °C failed.

## 2. Experiment

### 2.1. Materials

Tetraethyl orthosilicate (TEOS; ≥96%), ethanol (EtOH; ≥99.5%), ammonium hydroxide (NH_3_·H_2_O; concentration: 25%–28%), magnesium (Mg; particle size: 100 mesh–200 mesh; ≥99%), hydrochloric acid (HCl; concentration: 36.0%–38.0%) and hydrofluoric acid (HF; concentration: ≥40%) were purchased from Sinopharm Group Co., Ltd. (Shanghai, China). Conductive carbon black (denoted as Super P; purity: ≥99%) was obtained from SCM Chem., Shanghai, China; and sodium alginate was purchased from Aladdin Reagent Co., Ltd. (Shanghai, China) Nano-sized silicon (average size: 30 nm; purity: >99.9%) was purchased from HaoTian Nano Technology Co., Ltd. (Shanghai, China). Micro-sized silicon (size range: 2 μm–10 μm; purity: >99.9%) was donated by Fuzhou Sunout Energy & Material Technology Inc. (Fuzhou, China) All chemicals were used as received without further treatment.

### 2.2. Sample Preparation

Monodisperse silica (SiO_2_) solid spheres were synthesized using a modified Stöber method [18,19]. The as-prepared SiO_2_ spheres and magnesium powder (mass ratio of 1:1) were well mixed and placed in a tube furnace. The reaction temperature was set at 700 °C, 800 °C, or 900 °C starting from room temperature with a ramp rate of 5 °C min^−1^. The reaction was held for 16 h in a flowing Ar/H_2_ atmosphere (volume ratio: 95/5). The resulting product was thereafter soaked in 1 M of HCl solution for 6 h to remove the magnesium species followed by washing with deionized (DI) water and drying at 80 °C under a vacuum overnight. The samples treated with the magnesiothermic reduction for different temperatures and successive HCl washing were named 700-HCl, 800-HCl, and 900-HCl, respectively. Alternatively, the samples washed with HCl were further etched with a 4% HF aqueous solution for 1 h. These samples were indexed as 700-HF, 800-HF, and 900-HF, respectively. The names indicated the combination of different magnesiothermic reduction temperatures as well as different treatments.

### 2.3. Structure Characterization

The crystalline structure of the samples was determined via X-ray diffraction (XRD) using a Bruker (Billerica, USA) AXS D8 Advanced X-ray diffractometer (Cu *K*_α_; λ = 1.5406 Å) at 40 kV and 40 mA. The morphology and microstructure of the samples were characterized with a Hitachi (Tokio, Japan) S4800 field-emission scanning electron microscope at an accelerating voltage of 4.0 kV and a JEOL (Tokio, Japan) JEM-2100F transmission electron microscope (TEM) at 200 kV. The specific surface area measurement was performed using Micromeritics ASAP 2020 Accelerated Surface Area and Porosimetry system. The specific surface areas and pore size distributions were calculated using the Brunauer–Emmett–Teller (BET) theory and the Barrett–Joyner–Halenda (BJH) method, respectively.

### 2.4. Electrochemical Measurement

The working electrodes were fabricated with the as-prepared powders, sodium alginate binder, and Super P carbon conductive agent (mass ratio = 7:2:1), which were dispersed in deionized water and cast onto copper foil. The cast electrodes were dried in vacuum oven at 80 °C for 12 h and cut into discs with a 13 mm diameter thereafter. The average mass loading density of the silicon was controlled to be more than 1.0 mg cm^−2^ in each individual electrode except 900-HCl, for which we adopted a high mass loading density of 2.4 mg cm^−2^ in order to verify its feasibility for high-energy-density LIBs.

Coin-type (2032) half-cells were assembled in an argon-filled glove box (MBraun, Garching, Germany) using metallic lithium foil as a counter electrode and commercial Celgard-2300 film as a separator. Electrolyte obtained from Zhangjiagang Guotai-Huarong Chemical New Material Co., Ltd. (Zhangjiagang, China). was used, in which 1.0 M LiPF_6_ was dissolved in a mixture of ethylene carbonate (EC) and diethyl carbonate (DMC) (1:1 *v*/*v*) with 5 wt % of fluoroethylene carbonate (FEC). The electrochemical performance of the batteries was characterized with a multi-channel battery-testing system (LAND CT2000, Wuhan LAND electronics Co., Wuhan, China), which used charge/discharge galvanostatic cycling from 3.0 V to 0.005 V (vs. Li/Li^+^). The cycling performance test was carried out at a current density of 0.2 C for 200 cycles (1 C = 1000 mAh g^−1^). The rate performance was measured in the current density sequence of 0.1 C, 0.2 C, 0.5 C, 1.0 C, 2.0 C, 5.0 C, and 0.1 C (five cycles of each current density). The cyclic voltammetry (CV) scans and electrochemical impedance spectroscopy (EIS) data were collected on a electrochemical workstation (Solartron Analytical, 1470E). The CV test was carried out at a scan rate of 0.2 mV s^−1^ between 0.001 V and 3.0 V for 3 cycles. The EIS measurement was performed in the frequency range between 0.001 Hz and 1 MHz at an amplitude of 10 mV.

## 3. Results and Discussion

Figure 1a,b presents the XRD patterns of the pristine SiO_2_, the coarse product of the reduction, the HCl-washed sample 800-HCl, and the 800-HF sample. It can be seen that the pristine SiO_2_ exhibited a broad diffraction peak at 2θ of 22° indicating the amorphous state. After magnesiothermic reduction at 800 °C, the typical diffraction peaks of Si, MgO, and Mg_2_Si were obvious for the coarse product of the reduction. The reaction occurred as shown below:SiO_2_(s) + 2Mg(g) → Si(s) + 2 MgO(s)(1)
Si(s) + 2Mg(g) → Mg_2_Si(s)(2)

The subsequent treatment with diluted HCl selectively dissolved the MgO and Mg_2_Si byproduct, so only diffraction peaks of Si were obvious in the XRD patterns of 800-HCl. HF treatment did not change the crystallinity of 800-HCl.

Figure 1c–k show the morphology and crystallinity of the pristine SiO_2_ particles, 800-HCl, and 800-HF. The SEM and TEM images indicated that the SiO_2_ particles presented monodisperse spheres with an average diameter of 420 ± 6 nm (Figure 1c,d). The HRTEM image and SAED pattern in Figure 1e further proved the amorphous structure of the SiO_2_ particles, which was consistent with the XRD result shown in Figure 1a. The SEM image of the 800-HCl silicon exhibited a 3D continuous porous structure (Figure 1f) that was distinctly different from the structure of the pristine SiO_2_. The pore diameter as averaged by the SEM image was 325 ± 13 nm and the wall thickness was 113 ± 9 nm. Consistent with the SEM image, the TEM image also showed 3D continuous porous structures with a pore diameter of about 318 ± 15 nm (Figure 1g). The HRTEM image and SAED patterns indicated that the porous structure was composed of crystalline silicon (Figure 1h). In addition, the striking difference between the morphology of the 800-HCl and 700-HCl in that the mesoporous µm-sized particles were inter-connected with each other (Appendix A) indicated that the increasing magnesiothermic reduction temperature played an important role in the structure control of the silicon.

As shown in the SEM and TEM images in Figure 1i,j, no significant morphology change was observed after the HF treatment. Both the 800-HCl and 800-HF samples exhibited similar interconnected macroporous structures. Only the average pore size and wall thickness were modified slightly after HF etching (to 320 ± 14 nm (pore size) and 104 ± 11 nm (wall thickness), respectively). The slight increase in the pore diameter and decrease in the wall thickness were related to the removal of residual silica. We found that around 75% of the mass was retained after HF etching, which implied that the conversion yield from SiO_2_ to Si (percentage of reacted SiO_2_ in the total mass of pristine SiO_2_) via the magnesium reduction was 86.5%. However, considering that potential mass loss was possible during the repeated sample washing, etching, and centrifugation processes, the real conversion yield values could be even higher. The magnesiothermic reduction at 700 °C produced silicon with a yield of 63.7% (55% of mass loss after HF etching). This suggested that the reduction process was almost fully completed when the temperature was increased from 700 °C to 800 °C. The porous structure was stable enough to survive HF etching because the amount of the residual silica was limited after the magnesiothermic reduction at 800 °C.

The porosities of the 800-HCl and 800-HF were characterized using a nitrogen gas sorption measurement. As shown in Figure 1l,m, both samples exhibited a typical type-IV curve with an obvious hysteresis loop, demonstrating the presence of mesosized pores [20]. The BJH pore size distribution curves indicated that the main pore size was in the range below 50 nm, which further proved the mesoporous structure of the 800-HCl and 800-HF samples. We concluded that 3D hierarchical meso-/macrostructures existed in the two porous silicon samples by combining the SEM and TEM images and the pore size distribution profiles. The specific surface area and the pore volume of the 800-HF sample were slightly decreased due to the removal of residual mesoporous silica (88.4 m^2^g^−1^ vs. 108.3 m^2^g^−1^ and 0.26 cm^3^g^−1^ vs. 0.38 cm^3^g^−1^, respectively). Considering that 75% of the mass was retained and 25% unreduced silica was removed after HF etching, the specific surface area of the residual silica was calculated to be around 168 m^2^g^−1^ using the formula of (108.3–88.4 × 75%)/0.25. These results suggested that the residual silica made a significant contribution to the specific surface area of 800-HCl due to the existence of mesopores.

Based on the results addressed so far, the structure and morphology of the as-synthesized silicon was significantly modified when the magnesiothermic reduction temperature was increased from 700 °C to 800 °C [17]. A 3D macroporous network was formed directly after the magnesiothermic reduction at 800 °C. The detailed structure formation process of the porous silicon at 800 °C is illustrated in Figure 1. The Mg melted and the vapor reacted with silica from the outer surface to the inner core of the silica particles. The higher vapor pressure significantly promoted the reaction between the silica and magnesium. The in situ-generated silicon diffused together to construct a 3D continuous silicon network templated by the silica colloidal crystals. At 700 °C, the conversion from silica to silicon was limited because the temperature was only slightly higher than the melting point of magnesium (648 °C). A decent amount of silica still remained that kept the original spherical shape. The in situ-formed silicon on the surface of adjacent sphere particles diffused together and formed a continuous network as shown in the schematic structure in Figure 1. By removing the unreacted silica after the HF etching process, the 3D continuous porous silicon structure was uncovered. When the temperature was increased to 800 °C, the reduction of the silica was significantly enhanced because the magnesium vapor had a higher vapor pressure. Not only the surface, but also the interior regions of the silica nanoparticles were converted to silicon, leading to an almost total consumption of the silica particles. As a result, a 3D macroporous silicon network is generated directly after reduction as exhibited in the 800-HCl sample. Because there was only a limited amount of residual silica within the HCl-washed sample, the 3D continuous silicon network structure did not change significantly after HF etching. Only the specific surface area and porosity changed slightly due to the removal of the small amount of silica. 

The electrochemical properties of the 800-HCl sample were investigated. The cyclic voltammetry shown in Figure 2a indicated similar characteristic CV peaks belonging to typical crystalline silicon [21]. Two small peaks between 1.7 V and 0.5 V appeared in the first cathodic scan and disappeared in subsequent cycles. These two peaks were attributed to the formation of the SEI layer. The sharp peak at 0.1 V corresponded to the lithiation of the crystalline silicon [22] and the corresponding anodic peaks located at around 0.35 V and 0.52 V were due to the delithiation of the Li-Si alloys [23]. All peak currents in the CV curves gradually increased with increasing cycles due to an activation process of the electrode during cycling [24]. Meanwhile, the CV feature of the 800-HCl sample was almost the same as that of the 800-HF sample (Appendix A), which indicated that 800-HCl underwent the same fundamental electrochemical process as the HF-etched sample.

The typical galvanostatic discharge/charge curves of the 800-HCl sample are displayed in Figure 2b. A long flat plateau at around 0.1 V was observed in the first discharge cycle that corresponded to the lithiation of the crystalline silicon. In the initial charge curves, a plateau from 0.3 V to 0.5 V ascribed to the delithiation of the Li-Si alloys was also displayed. These discharge/charge profiles were consistent with the CV results. In detail, the 800-HCl electrodes achieved initial charge and discharge capacities of 1909 mAh g^−1^ and 2454 mAh g^−1^, respectively, with an ICE value of 77.8%. The irreversible capacity loss was ascribed to the formation of the SEI film and the capture of Li ions in active materials during the delithiation process [25]. Although the 800-HF electrode showed an increased initial charge capacity of 2259 mAh g^−1^ (Appendix A) due to removal of the residual silica, the ICE value of 76.3% was comparable to that of 800-HCl. The comparable ICE values originated from their similar structures and porosities, which indicated that the HF etching process did not modify the structure significantly. In addition, Appendix A shows that the capacity and ICE values of the 800-HCl were generally higher than those of 700-HCl (1562 mAh g^−1^/1084 mAh g^−1^, 69.4%), which was ascribed to the decreased amount of the residual silica and the decreased specific surface area [17,26]. We concluded that the lower capacity of the samples obtained without HF treatment could be improved by raising the reduction temperature. 

Furthermore, the cyclic performance of the 800-HCl sample was evaluated together with the 700-HCl sample as shown in Figure 2c. The 700-HCl sample delivered an initial charge capacity of 1084 mAh g^−1^, which decreased to 557 mAh g^−1^ after 200 cycles, corresponding to a capacity retention of 51%. Compared to 700-HCl, the 800-HCl porous silicon electrode exhibited a significantly increased cyclic capacity. It presented a charge capacity of 1909 mAh g^−1^ at the first cycle and a reversible capacity of 786 mAh g^−1^ after 200 cycles, which were higher than those of 700-HCl. The cyclic performance tests clearly indicated that the macro-/mesoporous structure significantly improved the cyclic performance of the 800-HCl anode compared to the commercial nano-sized and micro-sized silicon particles. The macropores effectively alleviated the mechanical stress generated by the volume expansion during lithiation and the continuous silicon wall acted as a lithiation host and mechanical support for the electrode. Furthermore, we also found that raising the magnesiothermic reduction temperature could increase the absolute reversible capacities because more silica was converted to silicon.

Compared with the HCl-washed samples, the samples after HF treatment presented further increased reversible capacities as displayed in Appendix A. After 200 cycles, the remaining charge capacity of the 800-HF electrodes was 1127 mAh g^−1^ with corresponding capacity retentions of 50%, which was higher than 800-HCl. However, the absolute capacity values of the 800-HF sample were similar to the reversible capacities of the 800-HCl electrodes calculated against the mass of the silicon only. This indicated that the HF etching treatment did not improve the cyclic stability of the porous silicon significantly, but mainly increased the absolute specific capacities by removing the unreacted silica species. These results further proved that the formation of macropores was a critical factor in achieving a good cyclic stability because the macropores could effectively accommodate the volume change and release the mechanical stress. The control of the conversion degree from silica to silicon via lifting the reduction temperature increased the absolute reversible specific capacities after 200 cycles.

Figure 2d depicts the rate performance of the 800-HCl and the 700-HCl samples. The 800-HCl electrode displayed reversible capacities of 1755 mAh g^−1^, 1517 mAh g^−1^, 1233 mAh g^−1^, 983 mAh g^−1^, 666 mAh g^−1^, and 79 mAh g^−1^ at the rates of 0.1 C, 0.2 C, 0.5 C, 1 C, 2 C, and 5 C, respectively. The electrode could almost regain its initial capacities at 0.1 C (1585 mAh g^−1^) when the current density was returned from 5 C, indicating a good cyclic stability. Considering that reasonable mass loading densities of more than 1 mg cm^−2^ were maintained by all the electrodes, the rate performance of the macro-/mesoporous silicon was good. In addition, the 800-HCl silicon presented a much better performance than the 700-HCl sample at 2 C because the capacity was increased by almost 40%. The specific surface area and pore volume ratio of the 800-HCl silicon were significantly lower than those of the 700-HCl sample. Thus, we concluded that the increased conversion of silica to silicon caused by raising the reduction temperature from 700 °C to 800 °C significantly enhanced the electrochemical kinetics and led to an improved rate performance at high current densities. The improved rate capability of the 800-HCl sample compared to the 700-HCl sample originated from its hieratical macro-/mesoporous structure feature, which effectively facilitated the electrolyte wetting and charge carrier transportation. Appendix A shows that the 800-HF sample delivered reversible capacities of 2228 mAh g^−1^ at a current density of 0.1 C and maintains 457 mAh g^−1^ at a high rate of 5 C. This suggested that the residual inactive silica deteriorated the rate performance of the HCl-washed samples because the electron/ion transportation was hampered. 

Figure 2e depicts the Nyquist plots of the 800-HCl and 700-HCl electrodes after 200 cycles. The electrochemical process is represented by the equivalent circuit model in the inset. The depressed semicircle in the high-frequency range stands for the charge transfer impedance (Rct) [27]. The Rct values of the 800-HCl and 700-HCl electrodes after 200 cycles were fit as 75.8 Ω and 82.7 Ω, respectively. Even though the BET surface area of 800-HCl was lower than 700-HCl, the Rct resistance of the 800-HCl electrode was still decreased, which indicated that the removal of residual silica made a higher contribution to the improvement in the electrochemical kinetics.

To confirm that the macroporous structure was good for the electrode integrity, the morphology change in the 800-HCl electrode after 200 cycles was characterized. It can be clearly seen in Figure 2f,g that the pristine electrode presented the macroporous structure of the 800-HCl samples. After the repeated lithium-ion intercalation/deintercalation processes, the porous morphology could be still maintained. These results indicated that the large number of pores in the 800-HCl electrode were beneficial to accommodating the volume change, leading to the stable structure and good electrochemical performance.

Further investigations were carried out to examine the possibility of tuning the structure and performance of the silicon with the reduction temperature increased to 900 °C. In addition, the mass loading density of the electrode was increased, which is essential in high-energy-density lithium-ion batteries [28]. A high mass loading density of 2.4 mg cm^−2^ was adopted for the 900-HCl sample, which was about twice that of the silicon electrodes explored in other previously reported works [28,29,30].

Like the 800-HCl sample, a 3D continuous macroporous structure was also observed for the 900-HCl sample as shown by the SEM and TEM images in Figure 3a,b. Both the HRTEM image and SAED pattern proved that crystalline silicon was also formed at 900 °C (Figure 3c). In addition, Appendix A shows that the specific surface area and pore volume were almost identical before and after HF etching (63.7 m^2^g^−1^ vs. 63.2 m^2^g^−1^ with both at 0.16 cm^3^g^−1^). This indicated that the silica nanoparticles were almost completely converted to silicon with a limited amount of residual silica species left in the 900-HCl sample. The conversion yield was calculated to be 91.8% according to the mass loss of 16% after HF etching, which was higher than that at 800 °C. 

The electrochemical performances of the 900-HCl electrode together with those of the 900-HF electrode are shown in Figure 3d–g. The cyclic voltammetry curves and the discharge/charge profiles of the 900-HCl electrode shown in Figure 3d,e were similar to those observed for the 800-HCl silicon shown in Figure 2c,d. Compared with 900-HF, 900-HCl presented a comparable cycle performance with an obvious electrochemical activation process appearing in the cycle that was caused by the high mass loading density of the active material [31]. The initial discharge/charge capacities of the 900-HCl sample were 1321 mAh g^−1^ and 1105 mAh g^−1^, respectively, reaching a maximum capacity of 1695 mAh g^−1^ and 1672 mAh g^−1^ after a gradual capacity increase [32,33]. After 200 cycles, the charge capacity of the 900-HCl sample was maintained at 885 mAh g^−1^, which was slightly higher than that of the 800-HCl silicon. Appendix A compares the cyclic performance of the porous silicon to those from previously reported works with high mass loading densities. The porous silicon anode presented comparable cyclic performance to the typical works listed in the table. It is worth pointing out that the silicon anodes listed in Appendix A were mainly composited with a carbon buffer medium, which was not the case for the porous silicon in this work. Thus, a further improved cyclic performance was expected if carbon composition was applied to the porous silicon. Figure 3g compares the rate performances of the 900-HCl and 900-HF electrodes. Although the 900-HCl electrode showed an inferior rate capacity due to its high mass loading density, it delivered capacities of about 2137 mAh g^−1^, 1715 mAh g^−1^, 1246 mAh g^−1^, 915 mAh g^−1^, 453 mAh g^−1^, and 16 mAh g^−1^ at the rates of 0.1 C, 0.2 C, 0.5 C, 1 C, 2 C, and 5 C, respectively. The capacity returned to 1792 mAh g^−1^ at 0.1 C when the current density was reversed back from 5 C. Considering that the mass loading densities of the electrode were more than 2 mg cm^−2^, the rate performance of the 900-HCl silicon was good.

To further evaluate the potential mechanisms underlying the different electrochemical responses of the samples obtained under different reduction temperatures without HF etching, a set of mechanical stress analyses upon lithiation of the 700-HCl and 800-HCl/900-HCl samples were performed. In these simulations, continuum mechanical models were constructed based on the representative volume elements (RVEs) that were extracted from the morphologies of 700-HCl and 800-HCl/900-HCl shown in Appendix A. Both silicon and silicon oxide are considered to be isotropic materials; the difference is that silicon expands and yields when lithium is inserted while silicon oxide does not. Accordingly, an expansion–yield constitutive relationship, which is commonly analogous to the classic thermal stress problem [34,35,36], was applied to silicon only. According to the results of the microcompression test by Korte et al., the yield stress of silicon is set as 7 GPa [37]. The partial molar volume and the saturation concentration of lithium are set as 4.92 × 10^6^ m^3^/mol and 3.65 × 10^5^ mol/m^3^, respectively [38]. The modulus of silicon and silicon oxide are considered to be 73 GPa and 40 GPa [39,40]. In addition, in order to consider the influence of the structure and avoid the stress difference caused by the concentration gradient, the lithium concentration should be evenly distributed.

According to the thermal stress theory in solid mechanics, it is known that when a structure is subjected to an increased thermal load, the materials in the structure will expand and deform, resulting in stress. This stress is derived from the nonuniformity of the thermal field and mechanical constraints. For a uniform lithium concentration field as discussed in the following cases, the inconsistency in stress within the 700-HCl and 800-HCl/900-HCl samples was mainly induced by their different morphologies. We observed that the maximum von Mises stresses in both structures rose linearly with the state of charge (*SOC*) and then reached plastic yield when they arrived at the yield limit. Since the volumetric content of silicon in the porous structure of 800-HCl was relatively higher, meaning that the lithiation deformation of the material under the same *SOC* would be greater, the silicon in this structure was hence more likely to yield during lithiation.

However, peak stress reaching the yield limit does not mean that all of the silicon within 800-HCl entered the plastic state and lost its ability to withstand the load. Thus, to identify the relationship between the structural morphology and potential structural failure, Figure 4 was also elaborated. The stress and plastic strain distribution were drawn under the same *SOC*. For the sake of comparison, the von Mises stress was normalized with the yield stress and the edges of the original structure were drawn with white lines. These results showed that the causes of stress and the underlying mechanical failure in these two structures were quite different. Due to the constraint of the central silica, the stress in the 700-HCl sample during lithiation mainly originated from the contact of adjacent shells. Attributed to the expansion of the silicon shell and the local effect of contact, the Si/SiO_2_ interface of this structure was twisted outwards from the extrusion zone, which could easily cause shear failure of the interface. Instead of this, the remaining silica material in the porous structure did not play the role of a mechanical restraint. Stress caused by silicon expansion was undertaken by its three-dimensional continuous structure. The deformation was directed into the hole during expansion, so the free surface of the porous structure was compressed and buckled due to the compression instability. Since the deformation was large, plastic deformation also surrounded these buckling areas. As the interface failure was irreversible, the cyclic performance for the 700-HCl sample was relatively worse in withstanding the alternate lithiation deformation. Therefore, from a mechanical point of view, although the 800-HCl/900-HCl sample yielded earlier during lithiation, it was more likely to maintain its mechanical integrity, resulting in a better electrochemical performance, which was also confirmed by the electrochemical cycle tests shown in Appendix A.

## 4. Conclusions

In summary, 3D hierarchical macro-/mesoporous silicon was successfully synthesized using a magnesiothermic reduction process with a reduction temperature higher than 800 °C and could avoid the etching treatment with the aggressive and highly toxic HF. The macro-/mesoporous structure was directly formed at 800 °C after HCl washing because the silica particles were almost fully converted to silicon by magnesium. The cyclic stability and rate performance was significantly improved in the porous silicon compared to the commercial nano-sized and micro-sized silicon due to the hierarchical macro-/mesoporous structure. The increase in the reduction temperature from 700 °C to 800 °C helped to improve the absolute reversible capacity and initial coulombic efficiency of the 800-HCl sample because more silica was converted to silicon together with a reduced specific surface area and an increased wall thickness. The rate performance of the 800-HCl silicon was also enhanced due to the decreased amount of the inactive silica with the increased reduction temperature. Further increasing the reduction temperature to 900 °C also generated macro-/mesoporous silicon with a higher yield and a good electrochemical performance with high mass loading densities. The mechanical stress analyses upon lithiation of the 800-HCl/900-HCl samples indicated that stress caused by silicon expansion could be undertaken by its 3D continuous structure, which facilitated maintenance of the mechanical integrity, thus leading to a better electrochemical performance. The mechanical stress analyses provided an effective principle in the structure design of high-performance porous silicon anodes.

## Data Availability

The data that support the findings of this study are available from the corresponding author upon reasonable request.

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
