# Peer review of "High-Temperature Magnesiothermic Reduction Enables HF-Free Synthesis of Porous Silicon with Enhanced Performance as Lithium-Ion Battery Anode"

_molecules, 2022, doi:10.3390/molecules27217486_

Round 1
Reviewer 1 Report
In this work, authors developed a facile HF-free method to prepare the porous silicon anode for high-performance Lithium ions battery. They have discussed the impact of reduction temperature from 700 to 900 degree on the properties of porous silicon. It is explored that the 3D continuous structure formed by the magnesiothermic reduction at 800℃ and 900 ℃ can undertake the mechanical stress effectively, which is responsible for improved cyclic stability compared to the silicon synthesized at 700℃. After carefully evaluating the paper, I would like to recommend “minor revisions” due to the following reasons.
1. In “Introduction”, the author claimed that the “…the corrosive HF etching process significantly increases the safety risk and makes it environmentally unfriendly…”, However, they use the HCl to treat the samples. As well known that the HCl is also harmful to the environment, how to explain this problem?
2. In terms of the sample preparation, why use the HF to further treat the sample after the HCl washing?
3. Regarding the BET analysis, the author claimed that the “the calculated specific surface area of the residual silica is around 168 m2g-1.” This makes me confusing. As I known, when we try to measure the specific surface area of the samples, the mass should be carefully measured and input to the analyzer, why do author use the mass based on the statement of “Considering that 75 % of the mass is retained after HF etching” ?
4. How about the LIBs performance of 800-HF?
